# The Newly Identified Trichoderma harzianum Partitivirus (ThPV2) Does Not Diminish Spore Production and Biocontrol Activity of Its Host

**DOI:** 10.3390/v14071532

**Published:** 2022-07-14

**Authors:** Rongqun Wang, Chenchen Liu, Xiliang Jiang, Zhaoyan Tan, Hongrui Li, Shujin Xu, Shuaihu Zhang, Qiaoxia Shang, Holger B. Deising, Sven-Erik Behrens, Beilei Wu

**Affiliations:** 1Institute of Plant Protection, Chinese Academy of Agricultural Sciences, No. 2 West Yuanmingyuan Rd., Haidian District, Beijing 100193, China; wangrongqun@163.com (R.W.); 820473469@foxmail.com (C.L.); jiangxiliang@caas.cn (X.J.); soldoros_tan@163.com (Z.T.); litoprrr@163.com (H.L.); xushujin626@163.com (S.X.); zshisnoone@163.com (S.Z.); 2College of Horticulture and Landscapes, Tianjin Agricultural University, Tianjin 300392, China; 3Key Laboratory for Northern Urban Agriculture of Ministry of Agriculture and Rural Affairs, Beijing University of Agriculture, Beijing 102206, China; shangqiaoxia@bua.edu.cn; 4Institute for Agricultural and Nutritional Sciences, Section Phytopathology and Plant Protection, Martin Luther University Halle-Wittenberg, D-06120 Halle (Saale), Germany; holger.deising@landw.uni-halle.de; 5Institute of Biochemistry and Biotechnology, Section Microbial Biotechnology, Martin Luther University Halle-Wittenberg, D-06120 Halle (Saale), Germany; sven.behrens@biochemtech.uni-halle.de

**Keywords:** *Trichoderma* spp., mycovirus, antagonism, biocontrol activity, chlamydospores, conidiation, plant development, plant pathogens

## Abstract

A new partititvirus isolated from a *Trichoderma harzianum* strain (T673), collected in China, was characterized and annotated as Trichoderma harzianum partitivirus 2 (ThPV2). The genome of ThPV2 consists of a 1693 bp dsRNA1 encoding a putative RNA-dependent RNA polymerase (RdRp) and a 1458 bp dsRNA2 encoding a hypothetical protein. In comparative studies employing the ThPV2-infected strain (T673) and a strain cured by ribavirin treatment (virus-free strain T673-F), we investigated biological effects of ThPV2 infection. While the growth rate of the virus-infected fungus differed little from that of the cured variant, higher mycelial density, conidiospore, and chlamydospore production were observed in the virus-infected strain T673. Furthermore, both the ThPV2-infected and the cured strain showed growth- and development-promoting activities in cucumber plants. In vitro confrontation tests showed that strains T673 and T673-F inhibited several important fungal pathogens and an oomycete pathogen in a comparable manner. Interestingly, in experiments with cucumber seeds inoculated with *Fusarium oxysporum* f. sp. *cucumerinum*, the ThPV2-infected strain T673 showed moderately but statistically significantly improved biocontrol activity when compared with strain T673-F. Our data broaden the spectrum of known mycoviruses and provide relevant information for the development of mycoviruses for agronomic applications.

## 1. Background

Mycoviruses infect members of all major fungal taxa and oomycetes [1,2]. So far, mycoviruses have been isolated from more than 100 fungal species, and certain types of these viruses infect multiple host species [3]. Several studies indicated that fungi and mycoviruses have remarkably complex relationships, and mycoviruses have been proposed as potential tools to further unravel fungal genetics and physiology [3,4]. In some instances, mycovirus infection of plant pathogenic fungi led to reduce virulence or even non-pathogenicity of the fungal host [5,6,7]. Therefore, understanding the effects of viral infection of fungi and the mechanisms by which viruses alter the phenotype of their fungal hosts is important and may also contribute to the development of new biological control strategies to combat fungal diseases in crops.

The genus *Trichoderma* (syn: *Hypocrea*) belongs to the class of Sordariomycetes in the phylum Ascomycota (https://www.fungaltaxonomy.org/; accessed on 20 June 2022). More than 375 *Trichoderma* species have been reported in various environments including soil, air, and plant surfaces [8]. Intriguingly, some *Trichoderma* strains exhibit potent antifungal activity, allowing us to employ these strains in biocontrol programs and possibly to reduce the impact of synthetic chemistries in plant disease control. In addition, *Trichoderma* strains have been reported to exhibit growth-promoting effects by manipulating plant hormone homeostasis, leading to the enhancement of photosynthesis and carbohydrate metabolism [9,10,11].

Both biocontrol and plant growth stimulating activities of *Trichoderma* strains may be strongly affected by mycovirus infections, which is a concern. However, research on mycoviruses infecting *Trichoderma* spp. is still in its infancy [12]. In the year 2009, unclassified double-stranded RNA elements were found in *Trichoderma* spp. isolated in the Chiang Mai province, China [13]. Interestingly, a relatively large number of 32 dsRNA viruses were identified in a collection of 315 *Trichoderma* strains isolated from contaminated artificial logs and sawdust bags used for cultivating the shiitake mushroom *Lentinula edodes*. The size and numbers of bands allowed the categorization of dsRNAs into 15 groups, and the sequencing of randomly selected cDNA clones suggested that two viruses belonged to the family of *Hypoviridae* [14]. Moreover, an unclassified dsRNA mycovirus isolated from *T. atroviride* NFCF028 was annotated as Trichoderma atroviride mycovirus 1 (TaMV1) [15]. Complete genome sequencing of dsRNA viruses from *T. atroviride* NFCF394 and *T. harzianum* NFCF319 identified Trichoderma atroviride partitivirus 1 (TaPV1) and Trichoderma harzianum partitivirus 1 (ThPV1) as novel mycoviruses of the family of *Partitiviridae* [16,17]. More recently, five novel mycoviruses, i.e., Trichoderma asperellum dsRNA virus 1 (TaRV1) [18], Trichoderma harzianum mycovirus 1 (ThMBV1) [19], Trichoderma harzianum mycovirus 1 (ThMV1) [20], Trichoderma harzianum hypovirus 1 (ThHV1) [21], and the new (+)ss RNA hypovirus Trichoderma harzianum hypovirus 2 (ThHV2) [22], were identified.

In this study, we have characterized a novel dsRNA virus isolated from a *Trichoderma harzianum* strain identified in a pool of 120 isolates collected at Xinjiang, Inner Mongolia and Heilongjiang, China. The complete genome of the virus was sequenced, and phylogenetic analyses based on an RNA-dependent RNA polymerase and a hypothetical protein identified this virus as a novel member of the *Partitiviridae*. Hence, the *T. harzianum* partitivirus was designated as ThPV2. Subsequent culture-based experiments revealed that this mycovirus affects *Trichoderma* colony morphology and increases the ability of the fungus to produce spores. Interestingly, despite ThPV2 infection, this *Trichoderma* strain showed considerable plant development-stimulating and biocontrol activities against several plant pathogenic fungi and an oomycete.

## 2. Methods

### 2.1. Extraction and Purification of dsRNA

In total, 120 *Trichoderma* isolates from the soil of Xinjiang, Inner Mongolia and Heilongjiang were collected from 2014 to 2016, using the five-points sampling method (Appendix A). All fungal isolates were cultured in potato dextrose (PD) liquid medium and incubated with shaking at 200 rpm at 28 °C for 2 days prior to dsRNA extraction. Mycelia were collected from liquid cultures by filtration (0.22-micron filters; Bingda company, Beijing, China) and stored at −80 °C. DsRNA was extracted and purified from mycelia using CF-11 cellulose column chromatography as described [23]. Isolated dsRNA was treated with RNase-free DNase I (TaKaRa, Dalian, China) and S1 Nuclease (TaKaRa, Dalian, China) following the manufacturer’s instructions [20]. The integrity of dsRNA was confirmed by electrophoresis on 1% (*w*/*v*) agarose gels.

### 2.2. RNA Sequencing

Prior to dsRNA sequencing, the quality of the isolated RNA was improved by purification with a RNAClean XP Kit (Beckman Coulter, Inc. Kraemer Boulevard Brea, CA, USA) and a Ribo-Zero rRNA Removal Kit (Epicentre, Madison, WI, USA). A total amount of >1 µg dsRNA was sequenced at the Biotechnology Corporation Shanghai, China, using high-throughput Illumina hiseq 2000 technology and Illumina Novaseq 6000 (Agilent Technologies, Santa Clara, CA, USA). Clean contig sequences were obtained by CAP3 EST and the NAR Databases About Viruses (https://digitalworldbiology.com/blog/bio-databases-2020-viruses-and-covid-19/nar-databases-about-viruses, accessed on 20 March 2020) were used for protein annotation. The sum of contig reads per kilobase per million mapped reads (RPKM) values were calculated for each species and based on the obtained annotations and the RPKM scores. Sequenced contigs with high homologies to mycoviruses were selected for further processing.

### 2.3. Characterization of the Mycovirus Genome

Based on the contig sequences, target fragments were amplified by RT-PCR and were cloned and confirmed by sequencing. Thus, with the exception of 5′- and 3′-termini, the genome sequences of RNA1 and RNA2 of the mycovirus from *Trichoderma* strain T673 were obtained. The missing 5′- and 3′-terminal end sequences of the viral RNA1 and RNA2 were obtained by 5′- and 3′-RACE [24,25]. Using Vector NTI Advance 11.5.4, contiguous sequences for single RNAs were assembled into the complete sequences of RNA1 and RNA2, yielding the complete viral genome [26].

### 2.4. Phylogenetic Analysis

The ORF finder tool (https://www.bioinformatics.org/sms2/orf_Find. HTML, accessed on 8 August 2020) was used to predict putative open reading frames (ORFs) [27]. The mycovirus sequences were used as queries in NCBI BLASTn searches, using the NCBI BLAST program (https://blast.ncbi.nlm.nih.gov/Blast.cgi, accessed on 10 August 2020) [28], to identify similar mycoviruses to be included in the phylogenetic analysis.

A phylogenetic tree was constructed using the maximum likelihood method in Mega 7.0 software (https://www.kent.ac.uk/software/mega-7, accessed on 10 August 2020) [29]. The phylogenetic tree was constructed using amino acid sequences of the RdRp-coding regions (11 mycovirus sequences) and of the coding regions of the hypothetical protein or CPs (10 mycovirus sequences), applying the best models of amino acid substitution of LG + G and LG + G + I + F, respectively (Mega 7.0 software; https://www.kent.ac.uk/software/mega-7, accessed on 10 August 2020) [29].

### 2.5. Mycovirus Elimination from Isolate T673

A ribavirin treatment of protoplasts of *T. hrzianum* strain T673 was used to establish an isolate cured from mycovirus infection. Strain T673 containing the mycovirus was cultured on potato dextrose agar (Beijing Suolaibao Technology Co., Ltd., Beijing, China) amended with 100 µM ribavirin (Biotopped Life Sciences Co., Ltd., Beijing, China) at 28 °C until sporulation. Spores were transferred to PD broth containing ribavirin (100 µM) and incubated with shaking (200 rpm) at 28 °C. After 18–20 h, the mycelium was collected by filtration and rinsed with sterile distilled water three times, followed by three washes with 0.6 M NaCl. The washed mycelium was protoplasted with lysing enzyme (20 mg/mL) from *Trichoderma* (BingDa Biotechnology Company, Beijing, China), as described [20]. Protoplasts were adjusted to a density of 10^3^ cells/mL in 1 M Tris-HCl, pH 8.2, containing 1.2 M sorbitol and 1 M CaCl_2_ (TSC) and allowed to regenerate at 28 °C for 2 days [20]. Individual colonies were selected and further cultured on potato dextrose agar (PDA) at 28 °C for 5 days. DsRNA was extracted and purified by CF-11 cellulose column chromatography, as described [23]. Strains lacking dsRNA were identified by electrophoresis, and the absence of dsRNAs was confirmed by RT-PCR. Cured strains were designated as T673-F (mycovirus-free).

### 2.6. Effects of Mycovirus Infection on the Antagonistic Activity of T. harzianum In Vitro

To evaluate the antagonistic activities of the mycovirus-infected and cured *T. harzianum* strains in vitro, six pathogens, i.e., Alternaria solanacearum, *Fusarium oxysporum* f. sp. *cucumerinum*, *Fusarium pseudograminearum*, *Botrytis cinerea*, *Rhizoctonia solani*, and *Phytophthora capsici*, were employed in dual confrontation assays on PDA [14].

### 2.7. Effects of Mycovirus Infection on the Biological Characteristics of T. harzianum

#### 2.7.1. Colony Morphology and Biomass

The strains T673 and T673-F were inoculated onto PDA, corn meal dextrose agar (CMDA), and Czapek–Dox agar (CZA; all from Merck, Darmstadt, Germany). The fungi were cultured in the dark at 28 °C, with 25 replicates per treatment. Mycelial growth rates were recorded daily, and differences in colony morphology were monitored. Furthermore, each of the strains were also inoculated into PD liquid medium and incubated at 28 °C with shaking at 180 rpm for 2 days. To compare differences in biomass accumulation between the two strains, the dry weights of mycelia were determined; 10 replications were performed with each strain. For statistical analyses, a *t*-test was used [30].

#### 2.7.2. Conidiospore and Chlamydospore Production

To identify the effects of mycovirus infection on *Trichoderma* conidiospore production, 5 μL of conidiospore suspensions of strains T673 and T673-F, adjusted to contain 1 × 10^7^ conidia per mL, were inoculated onto PDA and cultured in the dark at 28 °C for 7 days. Generated spores were suspended in 5 mL of sterile water and counted using a hemocytometer. Five μL of a T673 and a T673-F conidiospore suspension, containing 1 × 10^7^ conidiospores per mL each, were allowed to form chlamydospores in the dark at 28 °C for four and five days, respectively. The experiment was repeated three times, with eight plates each. Data were analyzed by *t*-test [30].

#### 2.7.3. Plant Growth Promotion and Biocontrol Activity

Cucumber (*Cucumis sativus* L. 9930) plants were used in these experiments. Strains T673 and T673-F were inoculated onto PDA plates and cultured in the dark at 28 °C for 5 days until sporulation. Cucumber seeds of similar size were soaked in sterile distilled water for 30 min, and disinfected by soaking in 1% (*v*/*v*) sodium hypochlorite for 5 min. After rinsing with sterile distilled water (3–5 times), sterilized seeds were first dipped into 2% (*w*/*v*) aqueous carboxymethyl cellulose (CMC) and subsequently dipped into spore suspensions containing 1 × 10^7^ conidia of strains T673 or T673-F per mL. The spore-dipped seeds were placed in a Petri dish with moistened sterile filter paper and cultured at 28 °C in an incubator (Zhejiang Jiangnan Pharmaceutical Machinery Co., Ltd., Hangzhou, China) for two days. After the seeds had germinated and developed cotyledons and roots, the cucumber seedlings were transferred to a test tube containing 10 mL of 1/8 MS medium and cultured in a greenhouse. Each variant of the experiment was conducted with 8 plants and repeated three times. The growth of cucumber seedlings was monitored at 13, 15 and 35 days post-treatment (dpt). Seedlings not treated with either *Trichoderma* strain were used as a control. The data analysis was performed by *t*-test [30].

In order to study the effects of mycovirus infection on *Trichoderma* biocontrol efficacy in cucumber against the causative agent of cucumber wilt, *F. oxysporum* f. sp. *cucumerinum* (*Foc*), plants were co-inoculated with strain T673 and *Foc* or strain T673-F and *Foc*. The inoculum was adjusted to contain 10^8^ conidia of *T. harzianum* and/or 10^7^ conidia of *Foc* per mL. Non-inoculated plants and plants inoculated with *Foc* alone served as negative and positive controls. Plants were grown in a growth chamber (Zhejiang Jiangnan Pharmaceutical Machinery Co., Ltd., Hangzhou, China) at 16:8 h light:dark and 23:18 °C rhythm. Each variant of the experiment consisted of 8 plants and was performed in three replicates. The growth and disease index (%age of dead cucumber plants) of cucumber seedlings were observed at 9 and 13 days post inoculation (dpi). Statistical differences were analyzed by *t*-test [30].

## 3. Results

### 3.1. Characterization of the dsRNA Mycovirus Genome

From the virus-infected isolate CTCCSJ-G-HB40673, designated as *Trichoderma harzinum* strain 673 (T673) (Figure 1A), two dsRNA fragments of approximately 1.7 Kb and 1.4 Kb were detected after treatment with DNase I and S1 Nuclease (Figure 1B,C).

High-throughput sequencing of the dsRNAs isolated from strain T673 yielded a total of 39,405 clean reads and six contigs. Appendix A shows the RPKM data for virus species that were identified by sequencing. The highest RPKM value (67,060.04309) was obtained for Ustilaginoidea virens partitivirus 2. Two contigs corresponded to the 1.4 k bp and 1.7 k bp fragments of the dsRNA from T673 (Figure 1 and Appendix A).

Primers were designed that matched the internal regions of the two contigs. RT-PCR was performed, and the amplification products designated as contigs 7 and 11 mycovirus cDNAs, respectively (Appendix A). RACE was employed to amplify the flanking sequences and to obtain full-length cDNA sequences. The primary cDNA and RACE sequences were assembled and confirmed the characterization of two dsRNAs (Figure 1C). The length of dsRNA1 (contig 11) is 1693 bp, and the length of dsRNA2 (contig 7) is 1458 bp, including the poly-A sequence motif.

To obtain clues as to whether dsRNA1 and dsRNA2 belong to the same virus, we sequenced the positive sense strands of both. Interestingly, the dsRNAs carry the same 5’ terminal sequence (GCCUUUUUGUCUCA), strongly suggesting the same origin of both RNAs. The sequence elements may serve as promoter recognition sites for the viral RNA-dependent RNA polymerase (RdRp) [20].

### 3.2. Organization of the T673-Derived Mycoviral Genome

As outlined above, our data suggest that the T673-derived mycoviral genome consists of two dsRNA fragments, i.e., dsRNA1 and dsRNA2 (Figure 1C). DsRNA1 contains an open reading frame (ORF1) spanning nucleotides 50 to 1624. BLASTp analyses predicted that dsRNA1 encodes a 524 aa RdRp (see also below). DsRNA2 contains ORF2, spanning nucleotides 166–1293, encoding a hypothetical protein of 375 aa (Figure 1D).

### 3.3. Phylogenetic Analysis and Taxonomy of Trichoderma harzianum Partitivirus 2 (ThPV2)

The amino acid sequences of the putative RdRp and the hypothetical protein derived from the mycoviral genome were used as queries for BLASTp searches to identify related mycoviruses (https://blast.ncbi.nlm.nih.gov/Blast.cgi: accessed on 20 August 2020). For the RdRp, the results revealed 14 closely related sequences, i.e., those of Colletotrichum gloeosporioides partitivirus 1 (76.4% similarity; accession QED88095.1) [26], Plasmopara viticola lesion-associated partitivirus 3 (72.8%; QHD64801.1) [31], Erysiphe necator-associated partitivirus 7 (72.6%; QJW70316.1) [31] and Plasmopara viticola lesion-associated partitivirus 4 (71.6%; QHD64807.1; see Appendix A for further information) [31]. BLASTp analyses of the hypothetical protein of the new mycovirus from *T. harzianum* revealed 12 homologous amino acid sequences of mycoviruses including, again, Colletotrichum gloeosporioides partitivirus 1 (63.6% similarity; QED88096.1) [26], as well as Erysiphe necator-associated partitivirus 7 (60.5%; QJW70323.1) [31], Phoma matteuciicola partitivirus 1 (61.6%; QDK65070.1) [32], Plasmoparaviticola lesion-associated partitivirus 4 (58.2%; QHD64811.1) [31] and Ustilaginoidea virens partitivirus 2 (54.6%; YP_008327313.1; see Appendix A for further information) [33] were identified.

In order to address the taxonomic classification of this mycovirus, we aligned the full-length protein sequences of the RdRp and the hypothetical protein with those reported for representative members of the genera Alphapartitivirus, Betapartitivirus, Deltapartitivirus, Cryspovirus, and Gammapartitivirus [34,35,36,37,38,39,40,41,42,43,44]. The resulting maximum likelihood (ML) phylogenetic tree, which was generated with both proteins by using the best-fit model of LG + G (with 1000 bootstrap replicates), further suggested that the mycovirus identified in strain T673 of *T. harzianum* belongs to the *Partitiviridae* family (Figure 2) and supports the results of the BLASTp searches.

In summary, considering the extensive homology of the RdRp of the virus isolated from *T. harzianum* strain T673 with the RdRp proteins of five other partitiviruses (>65% similarity), and of the hypothetical protein showing 63.9% similarity with Colletotrichum gloeosporioides partitivirus 1 [26], our data strongly suggest that the virus should be classified as a new member of *Partitiviridae*. As the virus was isolated from *T. harzianum*, we designated it as Trichoderma harzianum partitivirus 2 (ThPV2). The nucleotide and predicted amino acid sequences of dsRNA1 and dsRNA2 were submitted to NCBI and are available under the accession numbers OL457022 and OL457023 (Appendix A).

### 3.4. Biological Effects of ThPV2 on Trichoderma harzianum T673

#### 3.4.1. Generation of a Cured Strain

To study the effects of ThPV2 infection on the biological characteristics of *T. harzianum* such as colony morphology, biocontrol, and plant growth promotion, we cured strain T673 of ThPV2 infection by ribavirin treatment. RT-PCR analyses confirmed the absence of ThPV2 RNA in a representative cured strain designated as T673-F (Figure 3). This strain was used in the subsequent experiments.

#### 3.4.2. Effects of ThPV2 Infection on Morphology and Growth of *Trichoderma*

In order to identify effects of partitivirus ThPV2 on colony morphology, we compared the phenotype of strains T673 and T673-F on PDA, CMDA, and CZA plates (Figure 3A). On all media tested, the strain T673 grew more vigorously and produced more mycelium than strain T673-F. Interestingly, the pigmentation of strains T673 and T673-F differed. On PDA plates, strain T673 appeared dark green, whereas strain T673-F was yellow or light green, suggesting that this strain produced and secreted a yellow pigment on this substratum (Figure 3A). Figure 3B,C show RT-PCR data demonstrating the absence of ThPV2 from *T. harzianum* strain T673-F.

To test T673 and T673-F for differences in growth, we measured the increase in colony diameters per time unit on solid plates, as well as the fungal mass in liquid suspension using standard procedures. The data, as summarized in Appendix A, indicate no significant difference in growth rates between the two strains.

#### 3.4.3. Effect of ThPV2 Infection on Spore Production of *T. harzianum*

The production of conidiospores and chlamydospores is important for spreading and survival of *T. harzianum* in the environment. The latter is of particular importance for a fungus used in the biological control of plant pathogenic fungi. Therefore, we investigated the ability of strains T673 and T673-F to produce these distinct spore types (Figure 4). Interestingly, the levels of conidiospores and chlamydospores were highly significantly (*p* < 0.01) increased in strain T673, as compared to strain T673-F (Figure 4 and Appendix A). Thus, the infection of *T. harzianum* with ThPV2 is correlated with enhanced spore production (Figure 4).

#### 3.4.4. Effect of ThPV2 Infection on *Trichoderma*-Triggered Plant Development

*Trichoderma* species exhibit growth-promoting effects on plants. Thus, we examined whether the newly identified partitivirus ThPV2 affects plant development. To test this, we dip-inoculated cucumber seeds with conidial suspensions of the *T. harzianum* strains T673 and T673-F. Non-inoculated plants served as controls (Figure 5A). Intriguingly, both *T. harzianum* strains increased growth of stems and leaves, with more pronounced plant growth-promoting effects observed with the cured strain T673-F than with strain T673 (Figure 5B). Both *T. harzianum* strains significantly promoted the onset of flowering (Figure 5A,C and Appendix A).

#### 3.4.5. Effect of ThPV2 Infection on *Trichoderma* Antagonistic and Biocontrol Potential

As several synthetic fungicides are banned, the implementation of antagonistic microorganisms such as *T. harzianum* will gain importance in plant protection [45]. To analyze whether ThPV2 infection affects the antagonistic potential of *T. harzianum* against plant pathogens, strains T673 and T673-F were employed in confrontation assays with the ascomycetes *Alternaria solanacearum, Fusarium oxysporum* f. sp. *Cucumerinum*, *Fusarium pseudograminearum*, *Botrytis cinerea*, the basidiomycete *Rhizoctonia solani*, and the oomycete *Phytophthora capsici* (Figure 6A and Appendix A). Plates inoculated with the pathogens alone served as controls. Interestingly, the assays indicated only minor differences between strains T673 and T673-F in their ability to inhibit growth of the pathogens tested. However, though statistically not significant, strain T673 tended to show higher inhibition rates than strain T673-F against all tested pathogens, except for *Rhizoctonia solani* (Figure 6B and Appendix A). Of further interest was that inhibition zones between *Trichoderma* strains and plant pathogens were not observed, arguing against the secretion of antifungal secondary metabolites. Instead, after 14 days of co-cultivation, intense hyphal coiling was observed by light microscopy and taken as an indication of hyperparasitism (Figure 6C and Appendix A). Although the rate of hyphal parasitism is difficult to assess in a quantitative manner, the overall impression was that strain T673 exhibited more intense coiling of hyphae of the pathogens tested (Figure 6C; arrowheads) than strain T673-F.

Taken together, these data show that the bio-control fungus *T. harziannum* is able to hyperparasitize and inhibit pathogens belonging to distinct phyla, i.e., ascomycota, basidiomycota, and oomycota. Furthermore, infection with the partitivirus ThPV2 did not interfere with the antagonistic potential of *T. harzianum* in confrontation assays in vitro.

The effective control of a pathogen observed in in vitro confrontation assays may not necessarily indicate efficacy in a plant environment. We therefore explored the effects of ThPV2 infection of *T. harzianum* on the protection of cucumber plants against *F. oxysporum* f. sp. *cucumerinum* (*Foc*). Roots of cucumber seedlings were dip-inoculated with spore suspensions of *Foc*, as well as with spore suspensions of *Foc* combined with spore suspensions of *Trichoderma* strains T673 or T673-F. Non-inoculated plants served as controls. At 13 dpi, non-inoculated control plants had formed healthy cotyledons and primary leaves (Figure 7 and Appendix A). By contrast, all plants inoculated with *Foc* alone had collapsed and died. *Foc* infection also occurred in the presence of *T. harzianum* strain T673. However, some plants had survived and showed green leaves, indicating the plant-protective effect of the bio-control fungus. As compared with *T. harzianum* strain T673, the cured strain T673-F had a slightly reduced plant-protecting effect, as indicated by severe chloroses in all surviving leaves (Figure 7 and Appendix A).

In summary, we conclude that ThPV2 infection does not impair but may even stimulate the biocontrol potential of *T. harzianum*. These promising data demand assessment of the protecting activity of cucumber plants against natural *Foc* disease incidence occurring under field and/or greenhouse conditions.

## 4. Discussion

By the year 2020, the volume of pesticides applied worldwide has been estimated at approx. 3.5 million tons annually, with China as the major contributing country, followed by the USA and Argentina [46]. Due to concerns about the risks associated with the application of synthetic chemistries, authorities around the world have enacted legislation to reduce the use of pesticides in agriculture and in order to increase consumer and environmental safety [47]. On the other hand, reduced pesticide application would lead to an increased incidence of plant diseases and a significant reduction in potential yields [48]. Accordingly, in a scenario with lower use of synthetic chemicals, alternative disease control measures are required [45]. Among the most attractive alternative disease control measures are biological products such as antagonistic microorganisms, which have been used at increasing rates during the last decades [49,50]. Overall, biological control products had a market value of more than USD 3 billion in 2018 (https://www.gminsights.com/industry-analysis/biocontrol-agents-market; accessed on 20 June 2022), with *Trichoderma* species being the most commonly used biocontrol agents. Highlighting the importance of the genus *Trichoderma*, 30 strains belonging to distinct species are currently used in biocontrol products worldwide [50]. *Trichoderma* species with the strongest impact comprise *T. harzianum*, *T. atroviride*, *T. asperellum, T. polysporum*, and *T. viride* [47,48,51,52,53]. Remarkably, most, if not all, of the biocontrol strains are considered as generalists, as indicated by the enormous host range of economically important above- and below-ground asco- and oomycete pathogens such as *Fusarium oxysporum*, *Fusarium solani*, *Fusarium graminicola*, *Rhizoctonia solani*, *Botrytis cinerea*, *Colletotrichum acutatum*, *Sclerotinia sclerotiorum*, and several others [51,52]. The modes of action applied by *Trichoderma* species in combating pathogens include coiling and enzymatic lysis hyphae, the secretion of mycotoxic secondary metabolites, competition for resources, and, importantly, the stimulation of development and induction of resistance in the host [52].

Members of all major fungal taxa harbor viruses (for overview, see [2]), and this is also the case with different *Trichoderma* biocontrol species, including *T. harzianum* [12,13,14,20,50]. Infections with mycoviruses may cause hypovirulence or non-pathogenicity in plant pathogenic fungi [44,45,46,47,48]. At the beginning of the 20th century, the chestnut blight fungus *Cryphonectria parasitica* [53] virtually destroyed chestnut (*Castanea dentata*) tree populations in the US and in Europe, but, surprisingly, the occurrence of dsRNA viruses belonging to the *Hypoviridae* family caused a dramatic reduction in virulence in the virus-infected chestnut blight fungus and prevented the European chestnut from eradication [53,54]. The observation that the transmission of dsRNA elements through hyphal anastomoses conferred hypovirulence to non-infected fully virulent strains suggested novel biocontrol strategies against the chestnut blight fungus also in the US [54,55]. Accordingly, the observation that dsRNA viruses can strongly reduce the virulence of their fungal hosts [55,56,57,58,59] raised the question of whether mycoviruses in *Trichoderma* strains may affect the antagonistic activity of these biocontrol fungi as well as their plant growth-modulating activities [20].

In order to investigate whether dsRNA viruses capable of reducing virulence and biocontrol activity also occur in *T. harzianum*, we set out to identify virus-infected strains, to cure a strain from virus infection, and to compare the virus-infected and corresponding cured isogenic strain with respect to plant growth- and development-stimulating activities and corresponding antagonistic effects. From three dsRNA-harboring *T. harzianum* strains out of 120 that were identified in a previous study [60], we now report on their complete genome sequencing. Detailed phylogeny using the amino acid sequence of both putative viral proteins identified a novel dsRNA virus, designated as ThPV2. In this context, it is worth noting that the genomes of only eight mycoviruses, including ThPV2, have been sequenced worldwide to date. The other viruses are two mycoviruses from *Trichoderma atroviride* (Trichoderma atroviride mycovirus 1 [15,16] and Trichoderma atroviride partitivirus 1 [17]), four mycoviruses from *T. harzianum* (Trichoderma harzianum bipartite mycovirus 1 (ThMBV1) [19], Trichoderma harzianum mycovirus 1 (ThMV1) [20], Trichoderma harzianum hypovirus 1 (ThHV1) [22], and Trichoderma harzianum partitivirus 1 (ThPV1) [17]), and a single virus from *Trichoderma asperellum* (Trichoderma asperellum dsRNA virus 1 [18]). Of these, one mycovirus belongs to the *Fusagraviridae*, two belong to the *Partitiviridae*, one is a *Betahypovirus*, and three mycoviruses remain unclassified so far. In the present study, we show that ThPV2 is a newly identified member of the *Partitiviridae* family, expanding the database of known mycoviruses and our understanding of viral diversity.

In order to examine the effect of ThPV2 on plant development and biocontrol activity of *T. harzianum*, a partitivirus-free strain was generated. Interestingly, the virus-containing strain differed slightly from the cured strain in pigmentation and mycelial density and showed more efficient conidiation and chlamydospore formation. Chlamydospores are thick-walled resting spores formed to survive unfavorable conditions. Thus, the fact that formation of these structures is increased in the virus-infected *T. harzianum* strain is important, as it suggests that the antagonistic inoculum is likely maintained in agricultural environments. Intriguingly, neither the growth- and development-promoting activity nor the antagonistic potential against relevant plant pathogens, i.e., the ascomycetes *Alternaria solani, Fusarium oxysporum* f. sp. *cucumerimun, Fusarium*
*pseudograminearum*, and *Botrytis cinerea*, the basidiomycete *Rhizoctonia solani*, and the oomycete *Phytophthora capsici* were attenuated by ThPV2. Likewise, the previously compared virus-containing and virus-free strains 525 and 525F, respectively, of *T. harzianum* revealed that infection with the as yet unclassified dsRNA virus did not diminish the antagonistic potential against *F. oxysporum* f.sp. *cucumebrium*, *F. oxysporum* f. sp. *vasinfectum*, and *B. cinerea* [20]. Interestingly, the antagonistic activity against *Pleurotus ostreatus* and *Rhizoctonia solani* was higher in the partitivirus ThPV1-infected strain of *T. harzianum* NFCF319 than in the corresponding cured strain. As the virus-infected strain exhibited significantly increased β-1,3-glucanase activity, as compared with the cured strain, it can be hypothesized that ThPV1 may regulate β-1,3-glucanase activity and, in turn, virulence [17]. However, no increased β-1,3-glucanase activity was detected in *T. atroviride* infected by the *Alphapartitivirus* TaPV1 [16].

Distinct virus isolates, even if they belong to the same phylogenetic taxa, may differentially affect the phenotype of their host(s). We have completely sequenced the genome of a *T. harzianum*-infecting partitivirus and compared its effect on growth, sporulation, plant performance, and biocontrol efficacy of its host in detail. Although comparisons between the virus-infected and a cured strain indicated that ThPV2 did not cause adverse effects under defined conditions, it would be interesting to perform comparative greenhouse and field experiments with the ThPV2-infected and the cured strains. Such experiments could highlight environmental effects on the infection with the newly discovered partitivirus ThPV2. The data presented here expand the spectrum of known mycoviruses and provide further information for the development of mycoviruses for agronomic applications.

## Figures and Tables

**Figure 1 viruses-14-01532-f001:**
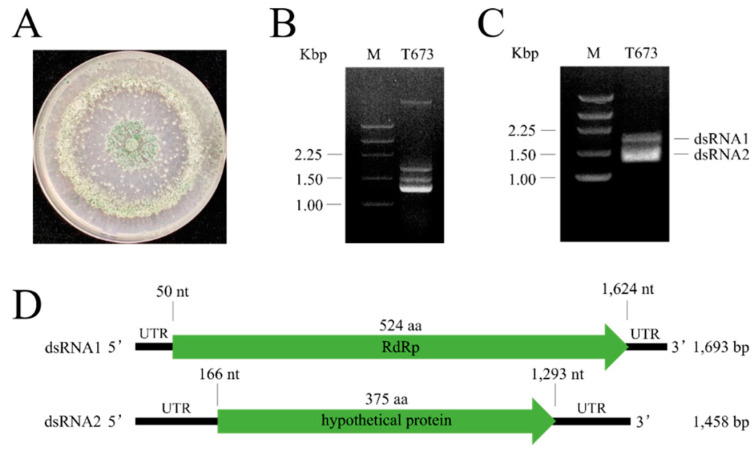
Identification of a dsRNA partitivirus in *T. harzianum* strain T673 and proposed genomic organization of the mycovirus. (**A**). Colony morphology of strain T673. (**B**). DsRNA extracted from strain T673. M: 250 bp DNA Ladder (TaKaRa, Kusatsu, Shiga, Japan); T673: dsRNAs from strain T673. (**C**). S1 nuclease- and DNase Ι-treated dsRNA from strain T673. M: 250 bp DNA ladder; T673. (**D**). Proposed genome organization of dsRNAs 1 and 2. ORFs are indicated as green arrows.

**Figure 2 viruses-14-01532-f002:**
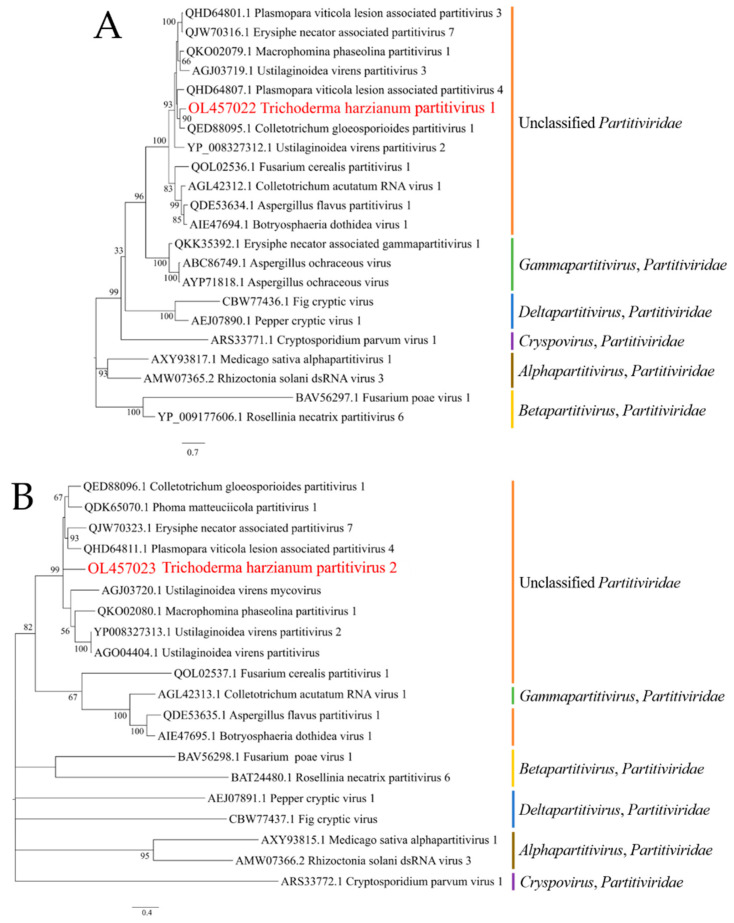
Maximum likelihood (ML) phylogenetic trees of viral RNA-dependent RNA polymerases (**A**) and hypothetical proteins with similarities to the hypothetical protein (**B**) of the novel dsRNA virus of *T. harzianum* strain T673. Both alignments suggest a close relationship with the *Partitiviridae* family (see text).

**Figure 3 viruses-14-01532-f003:**
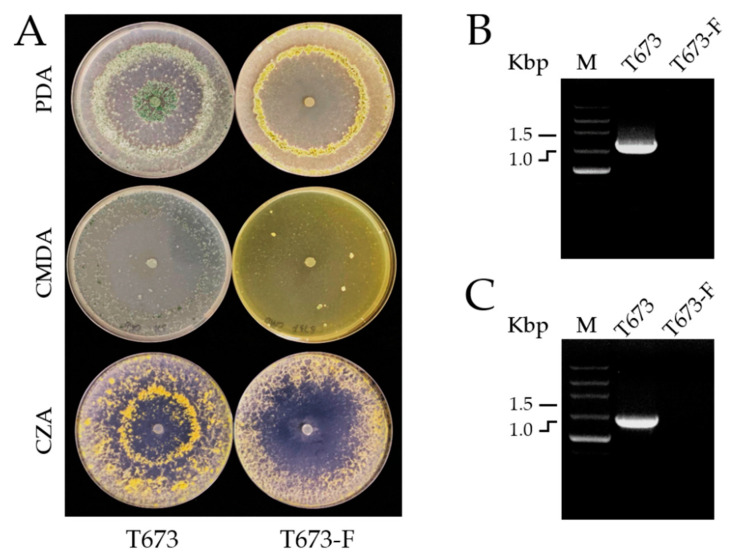
Characterization of ThPV2-infected and cured *Trichoderma harzianum* strains T673 and T673-F. (**A**): Colony morphology of strains T673 and T673-F on PDA, CMDA and CZA. (**B**): Amplification of a ~1.69 and (**C**): of a ~1.46 Kb ds-RNA genome fragments by RT-PCR in strain T673, but not in strain T673-F.

**Figure 4 viruses-14-01532-f004:**
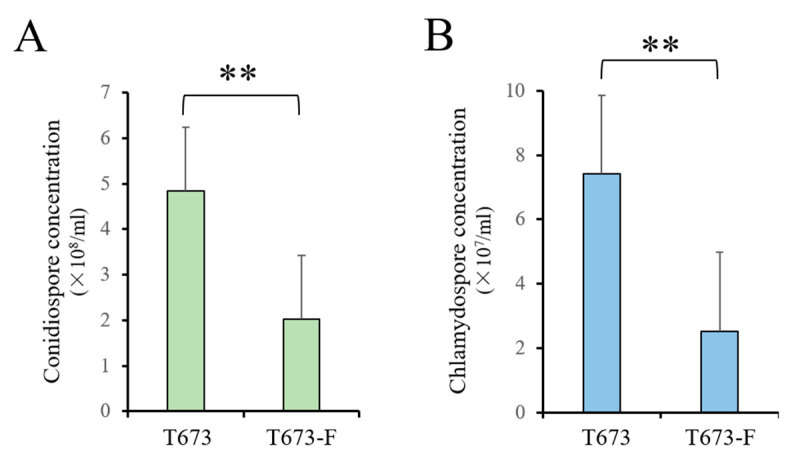
Conidiation and chlamydospore formation by strain T673 and partitivirus ThPV2-free strain T673-F. (**A**): Conidiospore formation was quantified on PDA at 5 dpi post inoculation. (**B**): Chlamydospore production was quantified on chlamydospore induction medium at 5 dpi. Double asterisks indicate highly significant differences (*p* < 0.01).

**Figure 5 viruses-14-01532-f005:**
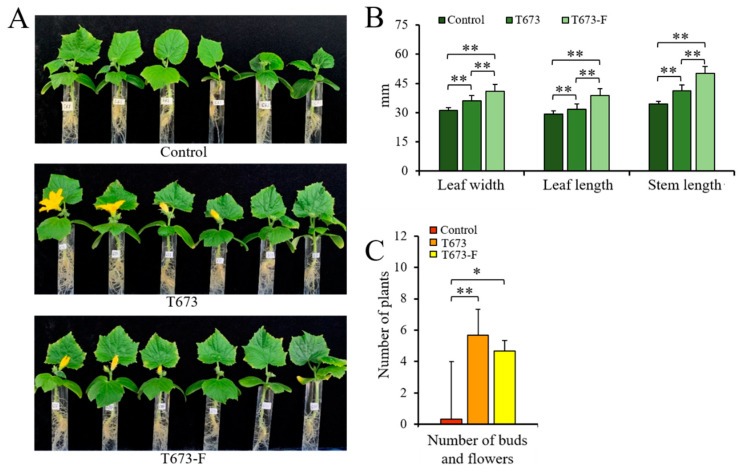
Effect of *T. harzianum* strains T673 or T673-F on cucumber plant development. (**A**): Phenotype of plants seed-dipped with *T. harzianum* strains T673 and T673-F. Non-treated plants served as controls. (**B**): Effect of treatment with strains T673 or T673-F on leaf width and length, as well as on stem length. Non-treated plants served as controls. (**C**): Effect of treatment with strains T673 or T673-F on budding and flowering of cucumber plants. Control plants remained non-inoculated. Every treatment (*n* = 8) was performed in three replicates. Data in (**B**,**C**) were taken from 35-day-old plants. *t*-test was used for statistical analysis. Single asterisks indicate significant differences (*p* < 0.05); double asterisks indicate highly significant differences (*p* < 0.01).

**Figure 6 viruses-14-01532-f006:**
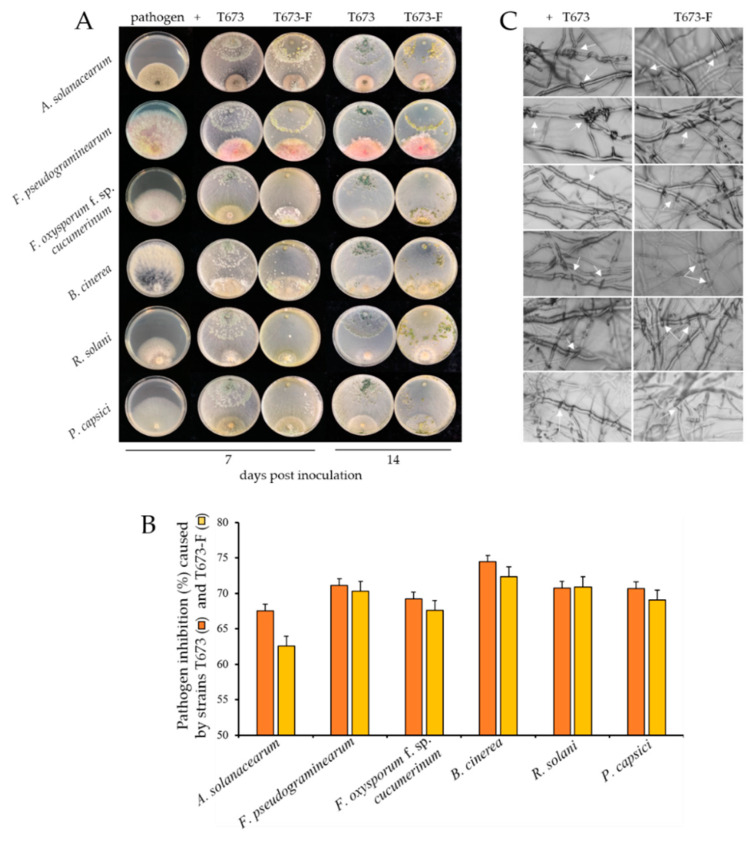
Antagonistic effects of *T. harzianum* strains T673 and T673-F against six plant pathogens. (**A**): Plate assays show that on PDA plates without the antagonist, the pathogens covered large areas of the plate at 7 dpi. When virus-infected or cured *Trichoderma* strains were co-inoculated with the pathogens, the area covered by the mycelia of the pathogens was clearly reduced. (**B**): Quantification of inhibitory effects of *T. harzianum* strains T673 and T673-F. Colony diameters were measured at 7 dpi. Bars indicate standard deviations. (**C**): Microscopy shows extensive coiling of hyphae of pathogens (arrows) by hyphae of ThPV2-infected and -cured *Trichoderma* strains in all interactions analyzed.

**Figure 7 viruses-14-01532-f007:**
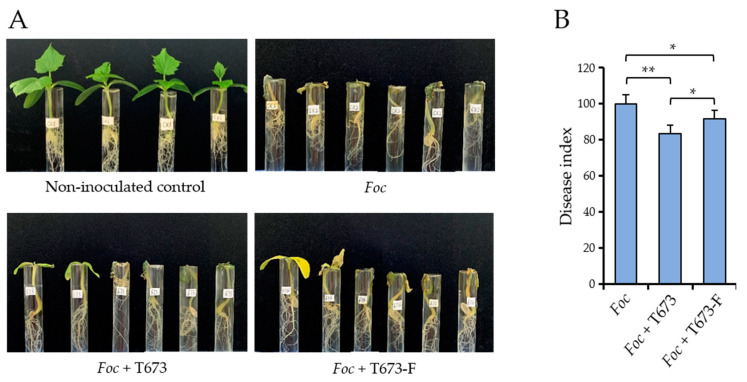
Biocontrol activity of *T. harzianum* strains T673 and T673-F against *F. oxysporum* f.sp.*cucumerinum* (*Foc*). (**A**): Phenotype of plants inoculated with *Foc*, or co-inoculated with *Foc* and *T. harzianum* strains T673 or T673-F. Non-inoculatd plants and plants inoculated with *Foc* alone served as controls. (**B**): Disease severity of plants inoculated with *Foc* alone or co-inoculated with *T. harzianum* strains T673 or T673-F, as indicated by the disease index (percentage of dead cucumber plants at 13 dpi). Every treatment (*n* = 8 plants) was performed in three replicates. Data in (**A**,**B**) were from plants at 13 dpi, and *t*-test was used for statistical analysis. Single asterisks indicate significant differences (*p* < 0.05); double asterisk indicates highly significant differences (*p* < 0.01).

## Data Availability

Not applicable.

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
