# Peer review of "The Newly Identified Trichoderma harzianum Partitivirus (ThPV2) Does Not Diminish Spore Production and Biocontrol Activity of Its Host"

_viruses, 2022, doi:10.3390/v14071532_

Round 1

Reviewer 1 Report

The manuscript by Wang et al. has improved since the previous submission. Nevertheless, there are still some minor issues to solve before acceptance.

L47: NCBI Taxonomy browser is not an authorized source for virus taxonomy, better cite the International Commission on the Taxonomy of Fungi (ICTF), https://www.fungaltaxonomy.org/

L74: “In this study we have characterized a novel …”

L217: “…we sequenced the positive sense strands of both dsRNAs. Interestingly, the dsRNAs carry the same 5’ terminal sequence (GCCUUUUUGUCUCA)...”

L512: “t-test”; a space between p, < and the value: “p < 0.05” “p < 0.01”

L526: “3 billion US$”

Author Response

Dear Ms. Zhang,

thank you again for the positive news. The minor modifications of our manuscript have now been made as requested by the reviewers. In detail, we respond as follows:

Response to reviewer 1:

L47: NCBI Taxonomy browser is not an authorized source for virus taxonomy, better cite the International Commission on the Taxonomy of Fungi (ICTF), https://www.fungaltaxonomy.org/

We have altered this as suggested. See also comment of reviewer 2 below.

L74: “In this study we have characterized a novel …”

We have corrected this as well.

L217: “…we sequenced the positive sense strands of both dsRNAs. Interestingly, the dsRNAs carry the same 5’ terminal sequence (GCCUUUUUGUCUCA)...”

We have corrected this as suggested by this reviewer as well.

L512: “t-test”; a space between p, < and the value: “p < 0.05” “p < 0.01”

A space has been inserted.

L526: “3 billion US$”

Modified as suggested.

With this, all authors thank you and, explicitly, the reviewers very much for your help.

We are looking forward to seeing the proofs and remain

With kind regards,

Beilei Wu

Reviewer 2 Report

lane 47. NCBI is not an authority source for nomenclature and classification. See www.speciesfungorum.org or catalogueoflive.org for relevant data. I suggest delete "sub-division Pezozomycotina" and mention "phylum Ascomycota".

There is typos on line 55 - stimulating

There is on line  - "...32 dsRNA viruses...", but "... isolates..." on line 63. I suggest used the term "viruses" there also.

Line 64 - should be ...of Hypoviridae.

line 114 - I suggest delete "genomes" . The single genome consists of RNA1 and RNA2 segments.

lines 147, 153 - complete line should be in Italic?

line 149 - six or five pathogens were employed?

line 155 - corn meal dextrose agar - why not "CMDA"?

Figure 2B - correct typos "Trichoderman harzianum"

line 544 - should be "Castanea dentata"

line 561- should be "detailed"

lines 562-564 - the sentence is not clear for me, maybe word is missing?? Which eight isolates you are talking about?

line 571 - should be "Betahypovirus"

Reference 7 is not in press recently, please correct.

Reference 9 - correct typos

Reference 25 - correct journal abbreviation

Reference 27, 31, 32, 47, 49, 52 - journal should be in Italics

Reference 30- correct typos Romero

Reference 35 - correct typos "profile"

Reference 45 - correct typos

Author Response

Dear Ms. Zhang,

thank you again for the positive news. The minor modifications of our manuscript have now been made as requested by the reviewers. In detail, we respond as follows:

Response to reviewer 2

lane 47. NCBI is not an authority source for nomenclature and classification. See www.speciesfungorum.org or catalogueoflive.org for relevant data. I suggest delete "sub-division Pezozomycotina" and mention "phylum Ascomycota".

In accordance with reviewer 1, we have cited cite the International Commission on the Taxonomy of Fungi (ICTF), https://www.fungaltaxonomy.org/. See comment to suggestion made by reviewer 1.

There is typos on line 55 – stimulating

We have corrected this.

There is on line  - "...32 dsRNA viruses...", but "... isolates..." on line 63. I suggest used the term "viruses" there also.

We have corrected this as suggested.

Line 64 - should be ...of Hypoviridae.

We have corrected this.

line 114 - I suggest delete "genomes" . The single genome consists of RNA1 and RNA2 segments.

We have deleted the word “genomes”, as suggested.

lines 147, 153 - complete line should be in Italic?

Not the complete lines should be in italics. The fungal names normally are given in italics, but in the headline, which is italicized, species names are not given in italics. This may depend on the editorial style, and if the technical editor should prefer to have the entire headline in italics, please feel free to change this.

line 149 - six or five pathogens were employed?

We apologize for this sloppiness. All six pathogens are included now.

line 155 - corn meal dextrose agar - why not "CMDA"?

We have changed that, CMD is now CMDA throughout the manuscript. Also in Fig. 3 we have changed this.

Figure 2B - correct typos "Trichoderman harzianum"

The figure has been corrected.

line 544 - should be "Castanea dentata"

This typo has been corrected.

line 561- should be "detailed"

This typo has been corrected as well.

lines 562-564 - the sentence is not clear for me, maybe word is missing?? Which eight isolates you are talking about?

We have modified this sentence, which now reads as follows: “… that the genomes of only eight mycoviruses, including ThPV2, have been sequenced worldwide to date.”

line 571 - should be "Betahypovirus"

This typo has been corrected as well.

Reference 7 is not in press recently, please correct.

We have added volume and page numbers.

Reference 9 - correct typos

We have added a space.

Reference 25 - correct journal abbreviation

Corrected according to Journal Abbreviation Database.

Reference 27, 31, 32, 47, 49, 52 - journal should be in Italics

We apologize for this sloppiness; the journal names are now in italics.

Reference 30- correct typos Romero

Corrected.

Reference 35 - correct typos "profile"

Corrected

Reference 45 - correct typos

We have corrected this as well.

With this, all authors thank you and, explicitly, the reviewers very much for your help.

We are looking forward to seeing the proofs and remain

With kind regards,

Beilei Wu